# N-palmitoyl-D-glucosamine, A Natural Monosaccharide-Based Glycolipid, Inhibits TLR4 and Prevents LPS-Induced Inflammation and Neuropathic Pain in Mice

**DOI:** 10.3390/ijms22031491

**Published:** 2021-02-02

**Authors:** Monica Iannotta, Carmela Belardo, Maria Consiglia Trotta, Fabio Arturo Iannotti, Rosa Maria Vitale, Rosa Maisto, Serena Boccella, Rosmara Infantino, Flavia Ricciardi, Benito Fabio Mirto, Franca Ferraraccio, Iacopo Panarese, Pietro Amodeo, Lea Tunisi, Luigia Cristino, Michele D’Amico, Vincenzo di Marzo, Livio Luongo, Sabatino Maione, Francesca Guida

**Affiliations:** 1Department of Experimental Medicine, Pharmacology Division, University of Campania “L. Vanvitelli”, 80138 Naples, Italy; monica.iannotta@unicampania.it (M.I.); belardocarmela85@gmail.com (C.B.); mariaconsiglia.trotta2@unicampania.it (M.C.T.); rosa.maisto@unicampania.it (R.M.); boccellaserena@gmail.com (S.B.); rosmainfantino@gmail.com (R.I.); flaviaricciardi93@gmail.com (F.R.); benitofabio.mirto@studenti.unicampania.it (B.F.M.); michele.damico@unicampania.it (M.D.); livio.luongo@gmail.com (L.L.); 2Institute of Biomolecular Chemistry (ICB) of National Research Council (CNR), 80078 Pozzuoli, Italy; fabio.iannotti@icb.cnr.it (F.A.I.); rmvitale@icb.cnr.it (R.M.V.); pamodeo@icb.cnr.it (P.A.); lea.tunisi@gmail.com (L.T.); luigia.cristino@icb.cnr.it (L.C.); vincenzo.dimarzo@criucpq.ulaval.ca (V.d.M.); 3Pathology Unit, Department of Mental and Physical Health and Preventive Medicine, University of Campania “L. Vanvitelli”, 80138 Naples, Italy; franca.ferraraccio@unicampania.it (F.F.); iacopo.panarese@gmail.com (I.P.); 4Canada Excellence Research Chair on the Microbiome-Endocannabinoidome Axis in Metabolic Health, Faculty of Medicine and Faculty of Agriculture and Food Science, Universitè Laval, Quebec City, QC G1V 0A6, Canada; 5I.R.C.S.S., Neuromed, 86077 Pozzilli, Italy

**Keywords:** N-palmitoyl-D-glucosamine, LPS, TLR4, cytokines, peripheral neuropathy, inflammation, mouse

## Abstract

Toll-like receptors (TLRs) are key receptors through which infectious and non-infectious challenges act with consequent activation of the inflammatory cascade that plays a critical function in various acute and chronic diseases, behaving as amplification and chronicization factors of the inflammatory response. Previous studies have shown that synthetic analogues of lipid A based on glucosamine with few chains of unsaturated and saturated fatty acids, bind MD-2 and inhibit TLR4 receptors. These synthetic compounds showed antagonistic activity against TLR4 activation in vitro by LPS, but little or no activity in vivo. This study aimed to show the potential use of *N*-palmitoyl-D-glucosamine (PGA), a bacterial molecule with structural similarity to the lipid A component of LPS, which could be useful for preventing LPS-induced tissue damage or even peripheral neuropathies. Molecular docking and molecular dynamics simulations showed that PGA stably binds MD-2 with a MD-2/(PGA)3 stoichiometry. Treatment with PGA resulted in the following effects: (i) it prevented the NF-kB activation in LPS stimulated RAW264.7 cells; (ii) it decreased LPS-induced keratitis and corneal pro-inflammatory cytokines, whilst increasing anti-inflammatory cytokines; (iii) it normalized LPS-induced miR-20a-5p and miR-106a-5p upregulation and increased miR-27a-3p levels in the inflamed corneas; (iv) it decreased allodynia in peripheral neuropathy induced by oxaliplatin or formalin, but not following spared nerve injury of the sciatic nerve (SNI); (v) it prevented the formalin- or oxaliplatin-induced myelino-axonal degeneration of sciatic nerve. SIGNIFICANCE STATEMENT We report that PGA acts as a TLR4 antagonist and this may be the basis of its potent anti-inflammatory activity. Being unique because of its potency and stability, as compared to other similar congeners, PGA can represent a tool for the optimization of new TLR4 modulating drugs directed against the cytokine storm and the chronization of inflammation.

## 1. Introduction

Chronic diseases, including neuropathic pain of various origins, are still scarcely managed in the clinic. Additionally, the mechanisms at the basis of neuropsychiatric symptoms associated with chronic pain are still poorly understood. Extensive evidence highlighted the role of inflammatory or immune system pathways in chronic pathologies including neuropathic pain [1,2]. Indeed, a critical role for proinflammatory mediators, as cytokines and chemokines from neurons and non-neuronal cells, together with a possible peripheral immune cells interaction, has been suggested in neuropathic pain pathophysiology [3,4].

As a key player of the innate immune response at the peripheral and central nervous system levels, the toll-like receptor 4 (TLR4) represents a target of alert for the inflammatory cells of an infectious or allostatic environment. TLR4 is able to bind the molecular patterns associated with pathogens (PAMPs), and thus recognizes lipopolysaccharide (LPS) derived from gram-negative bacteria (including those of the gut microbiota), as well as molecules of endogenous origin produced by cellular damage (DAMPs) [5].

TLR4 and other TLRs are therefore key sensors on which infectious and non-infectious challenges act with the consequent activation of the inflammatory cascade that has a role in acute and chronic diseases [6]. TLR4 is expressed by both peripheral and central nervous system cells including microglia and astrocytes [7,8]. TLR4 signaling is involved in the induction of tactile allodynia, which represents the main symptom associated with peripheral and central neuropathic pain [9,10]. Mice lacking TLR4 show significant reduction of both tactile allodynia and thermal hyperalgesia in the chronic constriction injury model of neuropathic pain [11]. However, in addition to driving inflammatory damage, a physiological tone of such receptors with protective function has recently been demonstrated in microglia [12].

*N*-palmitoyl-D-glucosamine (PGA) is a natural molecule that is structurally related to the lipid A component of LPS. It is produced by certain classes of bacteria such as *Rizhobium leguminosarum* [13]. Interestingly, bacteria of the *Rizhobia* genus are found in the mammalian gut microbiota, where they serve as xenobiotic metabolizers [14]. Previous studies have suggested the anti-inflammatory properties of PGA in animal model of inflammation and osteoarthritis pain [15,16], although the related mechanism of action has not been elucidated.

Due to the structural similarity of PGA with the lipid A component of LPS, we hypothesized that this molecule may modulate TLR4 activity, thereby exerting anti-inflammatory effects.

This hypothesis was confirmed by results obtained in the present study, where we investigated, through computational molecular docking and molecular dynamics simulations, the binding of PGA to the MD2 domain of TLR4, and its anti-inflammatory activity in LPS-stimulated murine macrophage-like RAW 264.7 cells and LPS-induced keratitis and corneal pro-inflammatory cytokine production. We have also studied the effects of PGA in various models of neuropathic pain, including LPS-induced decrease in pain threshold in the tail flick test and formalin- or oxaliplatin- or spared sciatic nerve injury (SNI)-induced allodynia in mice. Finally, PGA was tested against myelinoaxonal degeneration in these models of peripheral neuropathy as assessed by electron microscopy.

## 2. Methods

### 2.1. Computational Methods

Starting ligand geometry was built with UCSF Chimera 1.10.1 [17] and energy minimized with AMBER force field. The complexes underwent EM and then MD simulations with Amber16 pmemd.cuda module [18], using the ff14SB version of AMBER force field for the protein and lipid14 and glycam06 parameters for the ligand. To perform MD simulations in a solvent, the complexes were confined in TIP3P water periodic truncated octahedron boxes exhibiting a minimum distance between solute atoms and box surfaces of 10 Å, using the tleap module of the AmberTools16 package [18]. The full molecular dynamics protocol has been published elsewhere [19]. Production runs of MD simulations were carried out at constant temperature (300 K) and pressure (1atm) for 100 ns, with a time-step of 2 fs. Bonds involving hydrogens were constrained using the SHAKE algorithm [20]. The Cpptraj module of AmberTools16 and program UCSF Chimera 1.10.1 were used to perform MD analysis and to draw the figures, respectively.

### 2.2. Cell Culture and Luciferase Assay

RAW 264.7 (ATCC^®^ TIB-71™) cells were propagated in Dulbecco’s Modified Eagle’s Medium (DMEM) supplemented with 10% fetal bovine serum (FBS), 50 U/mL penicillin plus 50 μg/mL streptomycin, and 1% L-glutamine (Invitrogen, Milan, Italy), in a humidified atmosphere of 95% air/5% CO_2_ at 37 °C. HEK293 (Human Embryonic Kidney-293; ATCC Number: CRL-1573™) cells were propagated in Minimum Essential Media (MEM, cat. no. 31095-029 Invitrogen, Milan, IT) supplemented with 10% fetal bovine serum (FBS), 50 U/mL penicillin plus 50 μg/mL streptomycin, and 1% L-glutamine (Invitrogen, Milan, IT), in a humidified atmosphere of 95% air/5% CO_2_ at 37 °C. For cell transfection, RAW264.7 and/or HEK293 cells were seeded onto 24-well plastic plates at 2 × 10^3^ cells/cm^2^ density. After plating, the cells were transfected on the next day with: (a) pGL4.32[luc2P/NF-κB-RE/Hygro] Vector (Promega, IT; cat. no. E8491) and (b) a plasmid encoding for Renilla control vector (Promega, IT; cat. no. Cat. E2231) was used to normalize the signals and as positive control vector for monitoring transfection efficiency. The combination of plasmids was transfected into the cells by use of Lipofectamine LTX (Life Technology, MI IT) following the manufacture’s instruction. After 24 h, the cells were harvested to detect the luciferase gene reporter activity by the use of the Dual-Luciferase Reporter Assay System (Promega cat. E1910) and detected using the GloMax Luminometer (Promega).

### 2.3. RNA Extraction and Quantitative PCR (qPCR)

Total RNA isolation, purification, and cDNA synthesis from RAW264.7 and/or HEK293 cells were performed following published procedures [21]. Quantitative PCR (qPCR) was carried out in a real-time PCR system CFX384 (Bio-Rad) using the SsoAdvanced SYBR Green supermix (cat. n. 1725274, Bio-Rad Milan Italy) detection technique and specific primers. Quantitative PCR was performed on independent biological samples (*N* = 4). Also, each sample was amplified simultaneously in quadruplicate in a one-assay run with a non-template control blank for each primer pair to control for contamination or primer-dimer formation, and the cycle threshold (Ct) value for each experimental group was determined. A housekeeping gene (the ribosomal protein S16) was used to normalize the Ct values, using the 2^−^ΔCt^ formula; differences in mRNA content between groups were expressed as 2^^−ΔΔCt^, as previously described [21]. The primer sequences used: human TLR4 forward: GATAGCGAGCCACGCATTCA; human TLR4 reverse: TGGATTTCACACCTCCACGC; murine TLR4 forward: ATGGCATGGCTTACACCACC; murine TLR4 reverse: GAGGCCAATTTTGTCTTCCACA.

### 2.4. 3-(4,5-Dimethylthiazol-2-yl)-2,5- Diphenyltetrazolium Bromide (MTT) Assay

Cell viability was assessed by the MTT assay. The ability of cells to induce the reduction of MTT salts to formazan indicated the mitochondrial integrity and activity and has been interpreted as a measure of cell viability. Absorbance was read at 620 nm on a GENius- Pro 96/384 Multifunction Microplate Reader (GENios-Pro, Tecan, Milan, Italy). All compounds were dissolved in DMSO. Optical density values from vehicle-treated cells were defined as 100% of MTT-reducing activity and the treatment effects were measured as a percentage of the inhibition of control measurement [21].

### 2.5. TRPA1-HEK293 Cells and Calcium Assay

Stably transfected HEK293 cells with TRPA1 were generated, propagated and used for the calcium assay following published procedures [22].

### 2.6. Animals

Male CD1 (ENVIGO, Italy) weighing 25 g were housed three per cage under controlled illumination (12 h light/dark cycle; light on 6:00 a.m.) and standard environmental conditions (ambient temperature 20–22 °C, humidity 55–60%) for at least 1 week before the commencement of experiments. Mice chow and tap water were available *ad libitum*. The experimental procedures were approved by the Animal Ethics Committee of University of Campania “L. Vanvitelli,” Naples. Animal care was in compliance with Italian (D.L. 116/92) and European Commission (O.J. of E.C. L358/1 18/12/86) regulations on the protection of laboratory animals. All efforts were made to reduce both animal numbers and suffering during the experiment.

### 2.7. Induction of Keratitis

Intrastromal injection of 2 μg of Lipopolysaccharide (LPS, Sigma) from Pseudomonas aeruginosa, dissolved in 2 μL of phosphate-buffered saline (PBS, Sigma), was performed in both the eyes of male CD-1 mice under direct microscopic observation, according to previous studies [23,24,25]. Briefly, CD-1 mice were anaesthetized by pentobarbital (45 mg/kg in saline, i.p.) and tetracaine (1%) was topically applied on the ocular surface as a local anesthetic. To induce dilatation of pupils, tropicamide (5%) was instilled in each eye before the use of a 1/2-inch 30-gauge needle to make a nick into the corneal epithelium and anterior stroma. Then, a 10-μL gas-tight syringe with a 1/2-inch 33-gauge needle was introduced into the corneal stroma and advanced 1.5 mm to the corneal center to inject 2 μL of LPS solution or PBS alone.

#### 2.7.1. Treatments

CD-1 mice were randomized in the following experimental group (*N* = 5 each):mice receiving an intrastromal injection of PBS and oral vehicle administration (2 μL) (Saline/Veh group);mice receiving an intrastromal injection of LPS and oral vehicle administration (2 μg/2 μL PBS) (LPS/Veh group);mice administered *per os* with PGA (5 mg/kg) 30 min before receiving an intrastromal injection of LPS (2 μg/2 μL PBS) (PGA + LPS group);mice receiving an intrastromal injection of LPS (2 μg/2 μL PBS) and after 30 min administered per os with PGA 5 mg/kg for three days (LPS + PGA group);mice receiving an intrastromal injection of PBS (2 μL) and after 30 min administered per os with PGA 5 mg/kg for three days (Saline/PGA group).

At the end of the experiment, the keratitis clinical score was evaluated. Then, eyes were enucleated and corneas were removed as described by [26]. For each experimental group, *N* = 5 dissected central corneas were placed in cooled PBS, isolated from other ocular tissues, immediately frozen in liquid nitrogen and stored at −80 °C for subsequent biochemical analysis.

#### 2.7.2. Clinical Score Evaluation

An investigator unaware of the treatments evaluated the development of keratitis by using a biomicroscope. As previously described [24,27], keratitis was scored as follows: 0 = normal, clear cornea, no inflammatory reaction; 1 = mild corneal haze with visible iris; 2 = moderate corneal flare with moderate corneal haze and superficial punctate keratitis; 3 = significant corneal opacity, infiltration of cells in the stroma; and 4 = damage and loss of corneal tissue. Positive keratitis was scored as >1.

### 2.8. Cornea Immunohistochemistry

Corneas were paraffin-embedded and cut in 5μm serial sections [28], then the paraffin was removed by a xylene substitute (Hemo-De; Thermo-Fisher Scientific, Darmstadt, Germany). The immunohistochemistry procedure was performed as previously described [24] by BenchMark Automated IHC/ISH slide staining system (BenchMarkVentana, Tucson, AZ, USA). Sections were incubated with specific anti-ki67 (Rabbit polyclonal anti-Ki67 antibody; concentration 5 μg/mL; ab155807 Abcam) and anti-VEGF (Mouse monoclonal anti-VEGF antibody; 1:100; sc-57496, Santa Cruz) antibodies. After washes with PBE, the section was incubated with biotin-conjugated secondary antibodies and avidin-biotin-peroxidase complex (DBA, Milan, Italy). Specific antigens in each section were located with 3,3′diaminoenzidine (DAB) reaction, then slides were counterstained with hematoxylin. An expert pathologist analyzed the immunostaining (intraobserver variability 5%). Ki67 and VEGF antigenic expression was measured and calculated by Leica IM500 and statistics program Leica QWIN (Leica, Wetzlar, Germany). Four distinct preparations for each corneal sample were done and 18 fields of view were analyzed in each preparation for a total area of 1.35631e ± 0.003 μm^2^.

### 2.9. miRNA qRT-PCR

miRNA isolation from corneal homogenates was performed by using miRNeasy Mini Kit (Qiagen; Cat. No. 217004), according to ‘Purification of Total RNA, including Small RNAs, from AnimalCells’ protocol. In order to monitor the miRNA recovery efficiency and to normalize miRNA expression in the real-time PCR analysis, 5 μL of Syn-cel-miRNA-39 miScript miRNA Mimic 5 nmol/L (Qiagen; Cat. No. MSY0000010) was spiked into each sample, before miRNA isolation, in order to monitor the efficiency of miRNA recovery and to normalize miRNA expression in the subsequent real-time PCR (q-PCR) setting. Total RNA concentration was determined by NanoDrop 2000c Spectrophotometer (Thermo Fisher Scientific). miScript II RT(Qiagen; Cat. No. 218161) was used for the conversion of mature miRNAs to cDNA during the reverse transcription (RT) step. Mmu-miR-20a-5p, mmu-miR-106a-5p, mmu-miR-27a-3p, mmu-miR-206-3p, mmu-miR-381-3p levels were monitored by using SYBR Green PCR Kit (Qiagen; Cat. No. 218073) and specific miScript Primer Assays (Qiagen; Cat. No. MS00001309for mmu-miR-20a-5p; MS00011039 formmu-miR-106a-5p; MS00001351 for mmu-miR-27a-3p; MS00001869 for mmu-miR-206-3p; MS00032802for mmu-miR-381-3p and MS00019789 for Syn-cel-miR-39). In order to express miRNA levels as 2^^−ΔΔCt,^ ΔCt value was calculated as Ct miR target—Ct miR-39; then 2^^−ΔΔCt^ was obtained as 2^^−ΔCt^ of the treatment group/2^^−ΔCt^ of the control group [29].

### 2.10. ELISA Assay

Cytokines levels were assessed in corneal homogenates by using Mouse Cytokine Antibody Array, Panel A (ARY006, R&D System) according to the manufacturer’s instructions.

### 2.11. In Vivo Electrophysiological Recordings of Nociceptive Spinal Neurons (NS) in Combination with the Tail-Flick Test

On the day of electrophysiological recordings, mice were initially anaesthetized with Propofol (200 mg/kg i.p.) [30]. After laminectomy, spinal cord segments L4–L6 were exposed, and animals were fixed prone between the ear bars in a stereotaxic apparatus (David Kopf Instruments, Tujunga, CA, USA) and attached to the vertebral processes through clamps. The spinal cord was covered with mineral oil, to prevent the tissue from drying out. The single-unit extracellular activity of dorsal horn NS neurons was performed by using a glass-insulated tungsten filament electrode (3–5 MΩ) (FHC Inc, Bowdoinham, ME, USA) near the dorsal root entry zone, up to a depth of 1 mm [31]. Spinal neurons were defined as NS neurons when they were responding only to high intensity (noxious) stimulation [32]. In particular, to confirm NS response patterns, each neuron was characterized by applying mechanical stimulation to the ipsilateral hind paw using a von Frey filament with 97.8 mN bending force (noxious stimulation) for 2 s until it buckled slightly [33,34]. Only neurons that specifically responded to noxious hind paw stimulation were considered for recordings. The signal was amplified, filtered and displayed on a window discriminator; whose output was processed by an interface CED 1401 (Cambridge Electronic Design Ltd., UK) connected to iOS 5 PC.

Quantification of the neuronal activity was made off-line with Spike2 software (CED, version 5. The spontaneous and noxious-evoked neuronal activity was expressed as spikes/sec (Hz) and the effect of drugs was analyzed as firing rate, frequency, and duration of excitation. After recording a stable basal activity (15 min), intraperitoneal administration of vehicle or LPS (5 mg/kg, i.p.) was performed, and each recording was monitored until 240 min post-injection.

In particular, groups of animals were divided as following: 1) Naïve mice + Veh [100 µL of normal saline, 0.9% (*w*/*v*) NaCl] and 2) Naïve mice + LPS (5 mg/kg, i.p.). At the end of the experiment, each animal was killed with a lethal dose of urethane.

The tail-flick test was performed as described by [35] to assess behavioral responses to thermal noxious stimulus (radiant heat source) in mice. Briefly, using a tail-flick apparatus with an automated timer (Ugo Basile, Italy), the pain sensitivity was measured by tail-flick latency, which is defined as the time from the onset of radiant heat to tail withdrawal. A cut-off time of 10 s was set to prevent thermal injury.

### 2.12. Pain Models

*Formalin model.* Animals received formalin (1.25% in saline, 30 μL) in the dorsal surface of one side of the hind paw. Each mouse, randomly assigned to one of the experimental groups (*N* = 7–8), was placed in a plexiglass cage and allowed to move freely for 15−20 min. A mirror was placed at a 45° angle under the cage to allow full view of the hind paws. Lifting, favoring, licking, shaking, and flinching of the injected paw was recorded as nocifensive behavior [36]. The total time of the nociceptive response was measured every 5 min for 1 h and expressed in minutes (mean ± S.E.M.). After the acute evaluation, mice were tested in the long-term formalin-mediated pain behavior (see the section below) [37].

*Chemotherapy-induced peripheral neuropathy (CIPN).* Oxaliplatin 2.4 mg/kg (Sequoia Research Products, Pangbourne, UK) was dissolved in 5% glucose solution and administered intraperitoneally (i.p.) for 3 or 5 consecutive days every week for 3 weeks (13 i.p. injections). Control animals were treated with an equivalent volume of 5% glucose i.p. (vehicle). The dosage used mimics the clinical cumulative oxaliplatin dose causing chronic neuropathy [38,39,40].

*Spared nerve injury (SNI).* The spared nerve injury. Mononeuropathy was induced according to the method of Decosterd and Woolf [33,41,42]. Mice were anaesthetized by intraperitoneal injection of ketamine xylazine (60 mg/kg + 10 mg/kg). The sciatic nerve was exposed at the level of its trifurcation into sural, tibial, and common peroneal nerves. The tibial and common peroneal nerves were ligated tightly with 7.0 silk threads and then transected just distal to the ligation, leaving the sural nerve intact. Sham mice were anaesthetized, and the sciatic nerve was exposed at the same level, but it was not transected.

#### 2.12.1. Pain Measurement

*Von Frey Test*. Tactile allodynia was evaluated at a series of calibrated nylon Von Frey monofilaments (Semmes-Weinstein monofilaments, 2 Biological Instruments, Italy). Mice were allowed in the compartment of the enclosure positioned on the metal mesh surface and the test started after the mice were adapted to the testing environment for 30 min before any measurement was taken. The monofilaments, starting from the 0.008 g monofilament, was applied perpendicularly to the plantar surface of each hindpaw in a series of ascending forces (0.008, 0.02, 0.04, 0.07, 0.16, 0.40, 0.60, 1.0, 1.4 and 2.0,). Each stimulus was applied for approximately 1s with an interstimulus interval of 5 s. Withdrawal responses evoked by each monofilament was obtained from five consecutive trials. Voluntary movement, associated with the locomotion, was not counted as a withdrawal response. Tactile allodynia was defined as a significant decrease in the withdrawal threshold to the von Frey hair application. Each mouse served as its control, the responses being measured both before and after surgical procedures [37,43].

*Cold plate test.* The animals were placed in a stainless box (12 cm × 20 cm × 10 cm) with a cold plate as floor. The temperature of the cold plate was kept constant at 4 ± 1 °C. Pain-related behaviors (i.e., lifting and licking of the hind paw) were observed and the time (s) of the first sign was recorded. The cut-off time of the latency of paw lifting or licking was set at 60 s. [44].

#### 2.12.2. Treatments

Therapeutic effects of PGA treatment were evaluated in the different pain models. Mice received 5% pluronic acid (vehicle) or PGA at the doses of 2.5 5, 10 and 20 mg/kg (o.s.). Repeated treatments were performed ones a day and all the evaluations have been carried out before each injection. In the formalin model, animals were pretreated with PGA 30 min before subcutaneous formalin injection, and then one a day for one week. In the CIPN and SNI models, mice were preventive treated for 5 days before pain induction and then therapeutically treated for all the duration of the observation. PGA was kindly provided by Epitech group SpA (Saccolongo, Italy).

### 2.13. Electron Microscopy

Healthy and formalin-, oxaliplatinum- or SNI-induced neuropathic mice treated with vehicle or PGA (*N* = 3 per group) were deeply anesthetized, perfused transcardially with PBS (Phosphate-buffered saline, pH 7.4) and fixed with 3% paraformaldehyde/0.5–1% glutaraldehyde (vol/vol) in 0.1 M phosphate buffer (pH 7.4). Sciatic nerve sections (100-μm-thick) were treated with 0.5% OsO4 in PB for 30 min at 4 °C, dehydrated in an ascending series of ethanol and propylene oxide and embedded in TAAB 812 resin (TAAB). Ultrathin sections (50-nm thick) of the sciatic nerves were prepared and collected on Formovar-coated, single- or multiple-slot (50-mesh) grids and stained with 0.65% lead citrate. Morphometric analysis of sciatic nerves was performed at a magnification of 16500X by TEM microscope (FEI Tecnai G2 Spirit TWIN) in randomly selected fields of the nerve sections. After images acquisition, the myelinated profiles evaluable in the analyzed space were counted by two independent blinded observers and the histograms of quantitative distribution of normal vs. damaged/degenerated axon-myelinated profiles were generated according to the criteria previously described by Wozniak et al., 2018 [45]. Briefly, non-overlapping photographs were taken from the coronal nerve sections and an average of *N* = 50 microscopic fields per mouse (*N* = 3 mice per group) were analyzed to cover the entire cross-section through 2 mm length of the sciatic nerve at 5 mm from the proximal stump. MetaMorph software (Leica^©^) was used for quantitative ultrastructural analysis of the normal vs. damaged myelino-axonal profiles/micron2 of coronal section of the sciatic nerve.

## 3. Statistical Analysis

For cellular and biochemical analysis, the results of each experiment were reported as the mean ± S.D. or S.EM. To assess statistical significance among the groups, one-way ANOVA followed by Tukey Multiple Comparison test was used. Behavioral and electrophysiological data were expressed as mean ± S.E.M. Two-way ANOVA for repeated measures followed by Tukey’s post hoc test was used for comparison between groups. For the TEM analysis, the measure was expressed as means ± SEM of the number of myelin-axonal degenerated fibers/3micron2. One-way ANOVA followed Kruskal-Wallis test plus Dunn’s multiple comparison test was applied. The computer program GraphPad Prism version 6.0 (GraphPad Software Inc., San Diego, CA, USA) was used in all statistical analyses. A probability of *p* < 0.05 was considered sufficient to reject the null hypothesis.

## 4. Results

### 4.1. Theoretical Complex of MD-2 Protein with PGA

The activation of TLR4 by LPS is a complex and multistep process that require the binding of LPS monomers to LPS-binding protein (LBP), then to the cluster of differentiation 14 (CD14) and, ultimately, to the myeloid differentiation factor 2 (MD-2), which forms a complex with TLR4 on the cell membrane. The binding of LPS to MD-2 stabilizes the homodimeric complex (TLR4/MD-2/LPS)_2_, triggering the TLR4 signaling cascade. Thus, the binding to MD-2 is crucial for TLR4 responsiveness to LPS [46]. MD-2 features two β sheets arranged in a β cup fold, giving rise to a hydrophobic, open pocket, able to accommodate multiple LPS acyl chains, while the external rim of the pocket is lined by positive charged residues interacting with LPS negatively charged phosphate groups [47]. Due to its role in LPS recognition and binding, MD-2 represents the preferential target for hydrophobic ligands of TLR4 [48,49]. In this view, we decided to investigate the potential ability of PGA to bind this protein by a combined approach of molecular docking and molecular dynamics simulation. The solved x-ray structures of human MD-2 in complex with the antagonist lipid IVa (PDB id:2E59) and myristic acid (MA) (PDB id:2E56) shows that the alkyl chains of MA adopt the same conformation as the acyl chains of lipid IVa (Appendix A), suggestive of the occurrence of a preferred packing arrangement of the saturated alkyl chains in the hydrophobic pocket. Thus, rather than perform an automatic docking, we used the three C-14 chains of MA as a guide to manually dock the corresponding PGA molecules within the hydrophobic pocket and assessed the structural stability of the resulting MD-2/(PGA)_3_ complex by 100 ns of molecular dynamics (MD) simulations. For comparison purpose, we also simulated MD-2 in complex with two PGA ligands. A rmsd fluctuation analysis (Figure 1A–D) shows that both the whole ligands (Panels A, C) and the alkyl chains (Panels B, D) are by far more structurally stable in the MD-2/(PGA)3 than in the MD-2/(PGA)2 complex. Overall, the analysis of MD trajectory points out that PGA is able to stably bind MD-2 and that the preferred stoichiometry of the complex requires three ligands. This result is in agreement with the recent report showing that palmitic acid, a TLR4 agonist sharing the same saturated 16-carbon chain with PGA, binds MD-2 in the same protein/ligand ratio [50] of MA. As shown in Figure 1B, the alkyl chains are very stable during the simulated period and well fit on the alkyl chains of MA (Figure 1E), whereas the polar sugar units are endowed by higher mobility (Figure 1A–D). However, this is not surprising because the polar heads are exposed to the solvent, even though they engage stable polar interactions among each other and with residues lining the external rim of the pocket. In particular, H-bond interactions with an occurrence >10% over the whole MD trajectory are formed with Glu92 sidechain (>60%), the backbone of Val93 (>50%) and Ser120 (>50%) (Figure 1F).

### 4.2. PGA Prevents NF-kB Activation in LPS Stimulated RAW264.7 Cells

Following the computational study, we conducted a functional assay designed to assess nuclear factor (NF)-kappaB activation following the stimulation with LPS in murine macrophage-like RAW264.7 cells. In particular, RAW264.7 cells were transiently transfected with a plasmid (pGL4.32) carrying NF-κB response elements that once activated drive the transcription of the upstream luciferase reporter gene. Twenty-four hours after transfection, cells were pre-treated with various concentrations (up to 20 μg/mL) of PGA for 10 min and then stimulated with 100 ng/mL of LPS (+/− PGA) for 5 h (Figure 1G). As a control, we employed HEK293 cells that, in agreement with other studies [51], we found do not express TLR4 (Appendix A). Furthermore, the range of concentrations of PGA to use was determined in both RAW264.7 and HEK293 cells using the MTT cell viability assay (Appendix A).

Similar to previous studies [52], we found that LPS caused a robust increase in NF-κB activity in transfected RAW264 cells. Notably, the effect of LPS was prevented in the presence of the highest concentrations of PGA (10 and 20 μg/mL) (Figure 1G), whereas, in non-transfected RAW264.7 and/or HEK293 cells, the stimulation with LPS and/or PGA did not change NF-kB activity (Appendix A).

### 4.3. LPS-Induced Corneal Inflammation

#### 4.3.1. Clinical Score

After 3 days from intrastromal injections, all mice injected with LPS (2 µg) showed signs of keratitis. Particularly, severe alterations were evident in LPS corneal structure (clinical score: 3.9 ± 0.5; *p* < 0.01 vs. Saline/Veh). In contrast, PGA administration (5 mg/kg) after LPS attenuated the inflammation, as shown by the clinical score (2.48 ± 0.3, *p* < 0.01 vs. LPS/Veh). Interestingly, the maximum protection was reached when PGA was administered prior to LPS (clinical score: 1.74 ± 0.1, *p* < 0.01 vs. LPS/Veh) (Figure 2A,B) (*F* value (4, 10) = 50.65; *p* < 0.01).

#### 4.3.2. PGA Reduces ki67 and VEGF Labelling in LPS-Injected Mice

LPS intrastromal injection significantly increased the corneal staining of ki67 and VEGF (fold: +12 and +15 respectively, *p* < 0.01 vs. Saline/Veh) compared to mice injected with PBS. Specifically, ki67 was localized in the basal and suprabasal cell layers of the peripheral corneal epithelium, while VEGF staining was increased in the corneal stroma. PGA administration (5 mg/kg) did not affect ki67 and VEGF labeling in Saline/Veh group; on the contrary, a significant reduction in ki67 and VEGF protein levels (fold: −1.8 and −2.8 respectively, *p* < 0.01 vs. LPS/Veh) was observed in mice treated with PGA before LPS, when compared to LPS alone. The same effect was evident in mice treated with PGA following LPS intrastromal injection (fold: −1.5 and −2.7 respectively, *p* < 0.01 vs. LPS/Veh (Ki67: *F* value (4, 15) = 57.67; *p* < 0.01; VEGF: *F* value (4, 15) = 49.32; *p* < 0.01) (Figure 2C,D).

#### 4.3.3. PGA Affects TLR4-Dependent Cytokine Release through miRNA Regulation

Among the 5 miRNAs analyzed, miR-381-3p and miR-206-3p did not show any differential expression in our experimental setting (data not shown). Concerning the corneal levels of miR-20a-5p, miR-27a-5p and miR-106a-5p, they were not significantly affected by PGA administration (5 mg/kg) in mice receiving PBS intrastromal injection (Saline/PGA group), nor was TLR-4-dependent cytokine release (Figure 3A–C). On the contrary, LPS intrastromal injection (LPS/Veh group) significantly reduced corneal miR-20a-5p (fold regulation = −7.6, *p* < 0.01 vs. Saline/Veh) and miR-106a-5p (fold regulation = −30.5, *p* < 0.01 vs. Saline/Veh) (Figure 3A,C) expression. This effect was paralleled by a high increase of Il-6 (fold = +4.1, *p* < 0.01 vs. Saline/Veh), TNF-α (fold = +4.3, *p* < 0.01 vs. Saline/Veh) and Il-1β (fold = +2.7, *p* < 0.01 vs. Saline/Veh) levels (Figure 3D). Moreover, miR-27a-3p was significantly up-regulated in the LPS group (fold regulation = +8.3, *p* < 0.01 vs. Saline/Veh), with a marked decrement of corneal Il-10 content (fold = -2.9, *p* < 0.01 vs. Saline/Veh) (Figure 3D). PGA administration (5 mg/kg) following LPS intrastromal injection (LPS/PGA group) significantly increased miR-106a-5p levels (fold regulation = +9.9, *p* < 0.01 vs. LPS/Veh), while it reduced miR-27a-3p expression (fold regulation = −2.3, *p* < 0.01 vs. LPS/Veh) compared to LPS alone (Figure 3B,C). Moreover, a significant reduction of Il-6 corneal content was noticed in LPS/PGA mice (fold = −1.6, *p* < 0.05 vs. LPS/Veh), together with an increment of Il-10 levels (fold = +2.4, *p* < 0.05 vs. LPS/Veh). The same trend was shown by PGA/LPS mice (Il-6 fold = −1.5, *p* < 0.05 vs. LPS/Veh; Il-10 fold = +2.5, *p* < 0.05 vs. LPS/Veh). Interestingly, while miR-27a-5p was more reduced in mice receiving PGA pretreatment (PGA/LPS) compared to LPS/PGA mice (fold regulation = −3.5, *p* < 0.01 vs. LPS/PGA), PGA pretreatment did not modify miR-106a-5p expression compared to LPS/PGA mice. However, PGA/LPS mice were the only group to show a significant increment in miR-20a-5p levels compared to LPS group (fold regulation = +2.4, *p* < 0.01 vs. LPS/Veh). Also Il-1β and TNF-α were significantly reduced only by PGA pretreatment compared to LPS (fold = −1.9, *p* < 0.01 vs. LPS/Veh and *p* < 0.05 vs. LPS/PGA; fold = −1.9, *p* < 0.01 vs. LPS/Veh and *p* < 0.05 vs. LPS/PGA) (Figure 3D) (miR-20a-5p: F value (4, 25) = 21.86; *p* < 0.01; miR-27a-3p: *F* value (3, 19) = 34.74; miR-106a: *F* value (4, 25) = 29.44; *p* < 0.01; Il-1β: *F* value (4, 10) = 32.5; *p* < 0.01; Il-6: F value (4, 10) = 18.8; *p* < 0.01; Il-10: *F* value (4, 10) = 18.9; *p* < 0.01; TNF-α: *F* value (4, 10) = 52.0; *p* < 0.01).

Finally, LPS intrastromal injection significantly increased Il-1α levels compared to CTR group (fold = +4.1, *p* < 0.01 vs. Saline/Veh), while markedly decreasing Il-1rα levels (fold = −2.2, *p* < 0.01 vs. Saline/Veh) (Figure 3D). On the contrary, Il-1α levels were significantly reduced by PGA treatment (5 mg/kg) after LPS (fold = −1.6, *p* < 0.05 vs. LPS/Veh). This was paralleled by an increase of Il-1rα levels in LPS/PGA mice (fold = +1.6, *p* < 0.01 vs. LPS/Veh) The same trend was shown in PGA/LPS mice (Il-1α fold = −1.5, *p* < 0.05 vs. LPS/Veh; Il-1rα fold = +2.6, *p* < 0.01 vs. LPS/Veh) (Figure 3D). (Il-1α: F value (4, 10) = 61.3; *p* < 0.01; Il-rα: F value (4, 10) = 33.0; *p* < 0.01).

### 4.4. LPS Does Not Alter the Spinal Nociceptive Specific Neuron (NS) Activity, but Increases Responsiveness to a Thermal Stimulus in Naïve Mice

Based on LPS-induced inflammation, in vivo electrophysiological experiments were performed to investigate possible hyperexcitability in spinal nociceptive specific neurons (NS) after a single intraperitoneal administration of LPS (5 mg/Kg). The results were obtained from recordings of single NS neurons (one cell recorded from each animal per treatment) at a depth of 0.7–1.0 mm from the surface of the spinal cord (Figure 4A). This cell population was characterized by a mean rate of spontaneous firing of 0.015 ± 0.002 spikes/sec and only cells showing this pattern of basal firing were chosen for the experiment. In this study, we observed that a single intraperitoneal injection of LPS did not change the electrical activity of NS neurons in naïve mice, as compared to mice treated with vehicle (*p* > 0.99) (Figure 4B). In particular, NS neurons showed the following parameters: spontaneous activity (0.22 ± 0.20, spikes/sec), frequency (16.99 ± 0.18 spikes/sec) and duration (3.8 ± 0.0 sec) of noxious-evoked activity of NS neurons, 180 min post-LPS, as compared to the control group (0.10 ± 0.05 spikes/sec; 10.45 ± 2.61 spikes/sec; 3.87 ± 0.18 sec; respectively) (Figure 4C–E). These data, suggest the possible involvement of the activation of non-neuronal cells, such as microglia [53], in LPS-mediated spinal cord inflammation and, probably, that the LPS-preconditioning paradigm used in our study is not enough to globally activate microglia and in turn observe neuronal sensitization.

In this context, to assess if LPS administration might contribute to peripheral sensitization, we performed a tail-flick test in naïve mice. Differently from electrophysiological data, in the tail-flick test, naïve mice treated with LPS (5 mg/kg i.p.) showed a progressive reduction in latency of thermal responses that resulted significant at 180 min post-LPS (2.1 ± 0.25 s), with respect to baseline (4.75 ± 0.57 s, −15 min; *p* = 0.045) and naïve mice treated with vehicle (4.44 ± 0.52 sec; *p* = 0.0034) (Figure 4G). Considering the hypothesis of TLR4 activation involvement in the anti-nociceptive effect of PGA, we also tested the preventive effect of a single intraperitoneal injection of PGA at 5, 10 and 20 mg/Kg on tail flick latency in LPS-injected mice. In particular, tail-flick trials were evoked at 15 min intervals prior to the i.p. injection of vehicle or PGA at 10 or 20 mg/Kg, and regular intervals post-LPS injection (Figure 4G). Data showed that a single injection of PGA 20 mg/Kg (i.p.) effectively prevented systemic LPS-induced pain hypersensitivity in mice, 180 min post-LPS (4.1 ± 0.15 s, *p* = 0.0083), as compared to LPS-treated mice. On the contrary, no effect was observed after injection of PGA at 10 mg/Kg (i.p.) in LPS-treated mice (2.32 ± 0.12 s; *p* > 0.99) (Figure 4G). Two-way ANOVA analysis revealed a significant effect for time (*p* = 0.0020; F_7, 112_ = 4.41) and treatment (*p* = 0.0014; F_3, 16_ = 8.45), and significant interaction treatment X time (*p* = 0.0064; F_21, 112_ = 2.12).

### 4.5. PGA Prevented the Formalin- or Oxaliplatin-, but not SNI-Induced Allodynia

Formalin injection on the dorsal surface of hind-paw induced an increase in paw volume and spontaneous pain behavior reported as a typical biphasic response [54,55]. The first phase was characterized by a first robust nociceptive response followed by a transient decline thereafter. The second phase started 30 min after formalin reaching a peak at 50 min. A single PGA injection (5 and 10 mg/kg) prevented pain behavior in both phases of the formalin test in a dose dependent manner (time 35 min for PGA 5 mg/kg: 0.749 s ± 0.249 vs. time 35 min for Veh: 2.40 s ± 0.376; time 35 min for PGA 10mg/kg: 0.345 s ± 0.117 vs. time 35 min for Veh: 2.40 s ± 0.376, *p* < 0.0001, F_treatment (3,33)_ = 16.67; Two-way ANOVA followed by Bonferroni post hoc test) (Figure 5A). In agreement with our previous studies [37], we observed a significant decrease of the mechanical threshold in the injected and contralateral paw 7 days after formalin administration as compared to the saline-injected mice (Veh: 0.047 g ± 0.553 vs. BL 1.31 g ± 0.78 (Figure 5B). The subcutaneous injection of saline into the paw or oral vehicle administration did not change the pain response as compared with naive mice (not shown). We found that PGA (5 and 10 mg/kg) daily treatment significantly reduced the formalin-induced mechanical allodynia at 7 days in both the ipsi- and contralateral paws (PGA 5 mg/kg 0.55 g ± 0.459 vs. Veh: 0.047 g ± 0.553; PGA 10 mg/kg 0.76 g ± 0.612 vs. Veh: 0.047 ± 0.553; *p* = 0.0136, F _treatment (3,33)_ = 16.67; One-way ANOVA followed by Bonferroni post hoc test). Lower doses of PGA (2.5 mg/kg) did not exert any changes in acute or chronic treatment (Figure 5B).

The daily treatment of mice with a clinically relevant dose of oxaliplatin (2.4 mg/kg) induced an increasing painful condition. Indeed, on day 10, we found a reduction of threshold to mechanical, as well as cold stimuli which do not normally elicit pain. As measured by Von Frey apparatus, the mechanical withdrawal threshold was significantly lower in oxaliplatin-treated mice as compared with controls (Oxaliplatin/Veh 0.086 g ± 0.014 vs. Control: 1.040 g ± 0.256) (Figure 5C). Likewise, the licking latency in the cold plate test decreased from 0 sec to 30 sec (Oxaliplatin 16.40 s ± 3.50 vs. Control: 5.00sec ± 1.87) (Figure 5D). Remarkably, preventive administrations of PGA (20 mg/kg), starting from 5 days before oxaliplatin protocol, significantly (day 10 PGA 20 mg/kg 0.95g ± 0.102 vs. Control: 1.040 g ± 0.256; *p* = 0.0076 F_treatment (4.20)_ = 4.72; Two-way ANOVA followed by Tukey’s post hoc test; for mechanical allodynia) and (day 10 PGA 20mg/kg 18.60 sec ± 2.881 vs. Control: 16.40 s ± 3.50; *p* < 0.0001 F_treatment (4.20)_ = 13.65; One-way ANOVA followed by Tukey’s post hoc test; for cold allodynia) relieved pain in both tests (Figure 5D).

Three or seven days of SNI significantly reduced the tactile threshold as compared with sham animals (7 days SNI: 0.025g ± 0.012 vs. Sham 1.040g ± 0.256) (Figure 5E). Preventive (5 days) treatment with PGA (20 mg/kg) was not effective against SNI-induced allodynia (Figure 5E).

### 4.6. PGA Prevented the Formalin- or Oxaliplatin- but not SNI-Induced Myelino-Axonal Degeneration of Sciatic Nerve

The ultrastructural study of the sciatic nerve by electron microscopy confirmed our previous behavioral results. Formalin-, oxaliplatin-(CPN) or SNI-induced neuropathic mice showed severe myelino-axonal degeneration of the sciatic nerve evidenced by the presence of irregular shaped myelinated structures, differences in the thickness and alterations of the myelin sheath integrity in the cross-sections of the sciatic nerve (Figure 6A(c,g,e). These modifications were reverted or prevented by PGA injection in both formalin- or oxaliplatin-treated mice (Figure 6A(d,f)), but not in SNI-treated mice (Figure 6A(h)). The quantitative evaluation of the frequency of distribution of myelino-axonal degenerated profiles as a hallmark of damaged nerve revealed that PGA (5–20 mg/kg o.s.) was able to halve the frequency of distribution as several myelino-axonal degenerated profiles/micron2 of nerve section as reported in Figure 6B (mean number/μm^2^ ± SEM; CTL mice: vehicle: 0.51 ± 0.10/3 μ^2^ vs. PGA: 0.42 ± 0.09/3 μ^2^, *p* > 0.9999; Formalin-injected mice: vehicle: 4.33 ± 0.17/3 μ^2^ vs. PGA: 2.38 ± 0.14/3 μ^2^, *p* < 0.0001; CPN mice: vehicle: 4.98 ± 0.22/3 μ^2^ vs. PGA: 2.62 ± 0.18/3 μ^2^, *p* < 0.0001; SNI mice: vehicle: 4.24 ± 0.19/3 μ^2^ vs. PGA: 3.67 ± 0.21/3 μ^2^, *p* > 0.9999).

#### PGA Does Not Induce TRPA1-Mediated Elevation of Intracellular Ca^2+^

To understand whether PGA could activate TRPA1, we performed a calcium assay on HEK293 cells stably transfected with human TRPA1. Our experiments revealed that PGA (up to 10 µM) was ineffective in inducing TRPA1-mediated influx of Ca^2+^ ions in these cells or in counteracting the action of a canonical agonist of TRPA1 such us Allyl isothiocyanate (AITC) 100 µM (Appendix A).

## 5. Discussion

This study shows a possible mechanism of action of *N*-palmitoyl-D-glucosamine (PGA), a lipid molecule featuring a C-16 saturated alkyl chain bound through an amide bond to a D-glucose unit, by the direct binding to the MD2/TLR4 receptor complex. PGA resembles other TLR4 ligands based on a glucosamine core and linear carbon chains [56,57], and is produced by bacteria (i.e., genus *Rhizoma*) whose function is to facilitate the growth of other organisms (leguminous plants for example) or to promote mutual well-being between the host and its gut microbiome [13]. Molecular docking and molecular dynamics showed here that PGA stably binds MD-2 with a MD-2/(PGA)_3_ stoichiometry. Since palmitic acid has been shown to act as TLR4 agonist by direct binding to MD-2 [50] we can speculate that the glucosamine core of PGA is responsible for its switch towards antagonism. Therefore, to prove that this compound can antagonize the effect of LPS by acting on TLR4, we performed a luciferase assay in Nf-kB transfected RAW264.7 cells. Using this approach, we demonstrated that the Nf-kB activation triggered by LPS can be prevented by 10 and 20 µg/mL PGA. This effect, which was not found in HEK293 cells not expressing TLR4, corroborated our hypothesis that PGA has antagonist activity on this receptor.

Previous evidence showed the effectiveness of PGA in chronic inflammatory pathologic states like arthritis [15]. However, very little was known about the mechanisms of action of the PGA before the present study. PGA belongs to a group of mediators known *Autacoid Local Injury Antagonists* (i.e., the ALIA mechanism), a definition underlining their role as endogenous homeostatic factors produced on demand in damaged tissues to attempt tissue repair. This definition, however, does not provide any information on the molecular targets engaged by the PGA. This is the first study showing a relationship between PGA and the MD2/TLR4 complex, and the experiments performed here have also demonstrated the anti-inflammatory activity of PGA in pathologic conditions triggered by LPS, and hence involving TLR4 activation.

We first investigated PGA effects in a keratitis mouse model, by evidencing a significant reduction of keratitis score in mice receiving PGA. These effects on the molecular mechanisms due to persistent TLR4 activation induced by LPS were assessed by ki67 staining [58], which evidenced an increase in migration and proliferation of corneal cells after TLR4 activation by LPS, leading to a hyperplastic appearance of the epithelium [59,60]. Interestingly, ki67 positive cells were markedly reduced by PGA treatment. PGA effects were also evaluated on corneal neovascularization [61,62], usually occurring in inflammatory or infectious cornea diseases. Interestingly, PGA treatment led to a decrease of VEGF positive cells in corneal stroma, which are instead induced by LPS [63,64].

The anti-proliferative and anti-angiogenic effects of PGA were confirmed here by the reduction of pro-inflammatory corneal cytokine levels (Il-1α, Il-1β, Il-6 and TNF-α), paralleled by an increment in anti-inflammatory cytokines (Il-10, Il-1rα). Particularly, cytokine levels were modulated by miRNAs strictly involved in LPS response [62,65,66,67] and known to modulate TLR4-dependent cytokines, such as TNF-α and Il-β, targeted by both miR-20a-5p and miR-106a-5p [53,68]; Il-10, regulated by miR-27a-5p [67], and Il-6, modulated by miR-106a-5p [53] Overall, PGA was able to increase miR-20a-5p and miR-106a-5p, which were reduced by LPS intra-stromal injection. The increase of these miRNA by PGA was paralleled by the reduction of three important pro-inflammatory cytokines such as Il-6, TNF-α and Il-1β. On the contrary, PGA decreased miR-27a-5p levels, which exerts a pro-inflammatory effect by targeting Il-10. This anti-inflammatory cytokine, together with Il-1rα, was up-regulated by PGA treatments [53,68].

The findings from inflamed corneas correlate well with previous studies that described an anti-inflammatory and pain-relieving effect of PGA in arthritis or osteoarthritis models [15]. Chronic inflammation and the consequent functional changes of sensory fibers are traditionally distinct from those related to cell damage signaling (DAMP) or pathogens (PAMP) considered modulated only by immune cells. However, a close correlation between peripheral neural fibers and the immune system is now recognized, as in the case, for example, of the “sensory” functions of some previously considered purely immune-type cells (e.g., mast cells, microglia). Thus, it is not surprising that TLR4 activation drives the development and maintenance of neuropathic pain [4]. It is noteworthy that substances from Gram-negative bacteria, related chemically to LPS (i.e., LPS-RSU from *Rhodobacter sphaeroides*), were shown to block TLR4 and decrease neuropathic pain [69]. Consistently, we showed here for the first time that PGA, which is produced by bacteria of the genus *Rhizobia* [13], is a TLR4 antagonist and is capable of preventing both neuropathic pain and LPS-induced pro-inflammatory cytokines release.

There is, in fact, extensive evidence that innate immune signaling is necessary for the onset of pathological pain and that microglial cells play a key role for pain chronization [11,70]. To evaluate the pain-relieving effect of PGA, we first investigated the nociceptive threshold in the tail flick test after intra-peritoneal LPS in mice and, at the same time, recorded the nociceptive responses of spinal neurons. In agreement with previous data [71], no modification of neuronal activity during the four hours post LPS was observed. However, at the third hour post-LPS there was a reduction in the nociceptive threshold measured by the tail flick test. This effect, which was reversible after one hour, was prevented by PGA.

Since different phenotypes of neuropathic pain syndrome depend on varying factors, it is not unusual that therapeutic protocols in humans very often require combinations of pain relievers with different mechanisms of action to decrease allodynia, paresthesia and dysesthesia. We therefore investigated whether PGA was effective in three models of neuropathic pain, induced either by a chemical (e.g., oxaliplatin) or a peripheral nerve lesion (SNI). PGA produced a therapeutic effect only in the former case.

An unexpected finding in this study was that PGA, if administered for one week, was per se capable of lowering the nociceptive threshold under physiological conditions. On the contrary, allodynia was prevented by PGA in neuropathic mice. This might suggest a different role of TLR4 depending on whether healthy or chemically-induced neuropathic mice are treated. Based on new emerging findings of some unexpected roles for TLRs in neuroplasticity, it is possible that TLR4 blockade interferes with the tonic activity of immunity and may also alter the normal plasticity of spinal and supraspinal neural circuits. [49,72]. Further confirmation of an alternative protective role of the TLR4 receptor was obtained from a recent study in which deprivation of TLR4 signaling worsened neurodegeneration from viral infection. In that study, the authors showed that the homeostatic activation of microglia and their neuroprotective role depended on the intrinsic signaling of TLR4 [12]. Therefore, the allodynia observed after seven days of PGA treatment could be the consequence of a disruption of a fine functional balance between the immune and neural systems. In physiological conditions, both in utero or postnatal age and in adulthood, there is now clear evidence that TLRs perform complex functions including controlling neuroplasticity [48].

In a previous study we have shown that a single administration of formalin determines, apart from the classic noxious biphasic reaction, a progression towards a persistent peripheral neuropathy, which becomes apparent (with allodynia and hyperalgesia) a few days later and then lasts for several weeks [73,74,75]. We found that after single intra-paw formalin injection, PGA was able to decrease the first phase and completely prevented the second phase of the nocifensive behavior in mice. The fact that the first phase was also in part decreased could suggest that PGA might act as an indirect inhibitor of the TRPA1 channel [76,77,78], apart from to directly inhibit the TLR4 signal transduction. However, our data demonstrate that PGA does not activate TRPA1 receptor. Furthermore, we found that PGA was effective at reducing all pain symptoms in mice which developed neuropathy with allodynia and hyperalgesia one week after formalin injection.

Likewise, peripheral nerve damage and the associated pain symptoms caused by administration of the antineoplastic agent, oxaliplatin, were significantly reduced by PGA administered prior to oxaliplatin. The mechanisms underlying the neurotoxicity induced by chemical agents are diverse and include damage in the assembly and functioning of microtubules within the nerve fibers, with subsequent severe axonal toxicity and deficit of neuro-mediator transport. Regarding the bio-molecular or receptor mechanisms that could explain the very similar pathophysiological state induced either by oxaliplatin or by formalin, but not by SNI, these could lie in the fact that both cytoskeletal toxins and aldehyde products, as well as LPS, are powerful activators of TRPA1 receptors expressed in both nerve fibers and cells that participate in neuro-immuno-inflammation [76,79]. Indeed, thought PGA was not capable to activate TRPA1 in our in vitro system, we cannot exclude a possible indirect interaction between TLR4 and TRPA1 [80]. In fact, although the cellular and systemic effects induced by LPS are mainly attributed to the activation of the TLR4 signaling cascade, recent evidence demonstrates that this bacterial toxin activates several members of the TRP channel family, including TRPA1 [81].

It is very likely that PGA, by acting as a natural inhibitor of TLR4, can prevent the innate immunity response and peripheral cytokine release, which participate in neuronal sensitization and nerve inflammatory reactions. In fact, PGA reduced myelin damage only if this was generated by formalin or oxaliplatin, but not if the axons were sectioned entirely (SNI). Since PGA counteracts the excessive activation of cell-mediated peripheral neuro-immune responses, which lead to nerve demyelination, this suggests that this compound is effective only when axon integrity is maintained, and hence acts selectively on peripheral cells coating the axons and ensuring their normal functioning. This also leads to hypothesize the possible use of PGA in central neurological diseases of the demyelinating type, such as multiple sclerosis, where the primary damage is to the non-neuronal cells responsible for myelination.

The anti-allodynic effects of PGA were also supported by anatomical observations conducted by ultrastructural analysis (TEM) on the sciatic nerve in different mouse models of neuropathy. Electron microscopy revealed a significant reduction in the frequency of myelin-axonal degenerated profiles/micron^2^ of sciatic cross-sectioned area in the case of both intra-paw formalin-injected mice and in mice receiving systemic oxaliplatin, in comparison to SNI-treated mice, where PGA was ineffective.

In conclusion, we have shown how a molecule of bacterial origin, probably produced also by the human intestinal microbiome, can exert anti-inflammatory and immune-regulatory effects through the direct binding on the MD2 protein. Additionally, we have highlighted that PGA is also capable of preventing local or systemic damage induced by LPS and, in particular, peripheral nerve damage induced by antineoplastic therapy. Future studies may be important to characterize other related natural molecules and provide us with new drugs directed against TLR4 in controlling the cytokine storm in chronic inflammatory syndromes.

## Figures and Tables

**Figure 1 ijms-22-01491-f001:**
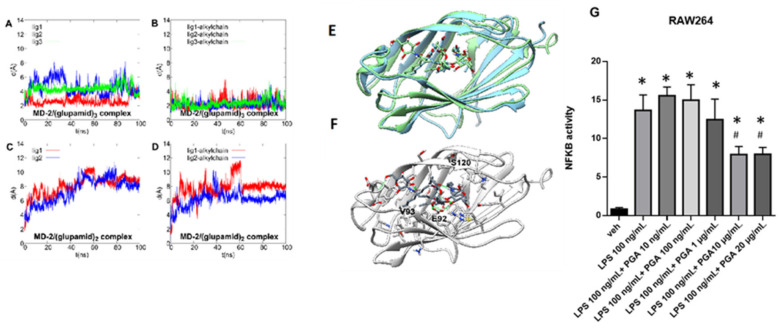
Root mean square deviation (rmsd) plot of PGA ligands during 100 ns of MD after protein backbone best-fit for whole ligand in MD-2/(PGA)3 (**A**), ligand alkyl chain only in MD-2/(PGA)3 (**B**), whole ligand in MD-2/(PGA)2 (**C**) and ligand alkyl chain only in MD-2/(PGA)2 (**D**) complexes. (**E**) best fit at level of protein backbone between x-ray structure of MD-2 in complex with myristic acid (PDB id: 2E56), colored in sky blue, and the representative MD frame of MD-2/(PGA)_3_ theoretical complex, colored in light green. (**F**) representative MD frame of MD-2/(PGA)_3_ theoretical complex. Protein is colored in light gray while PGA molecules in dark gray. Oxygen, nitrogen, sulfur and polar hydrogen atoms are colored in red, blue, yellow and white, respectively. PGA molecules and residues within 5 Å from the ligands are shown in stick representation. H-bonds are shown as green springs. (**G**) Luciferase activity measured in relative light units (RLU) normalized to the vehicle group was measured in NF-kB transfected RAW264 cells stimulated with LPS in the presence or not of the indicated concentrations of GPA. Values are mean ± SEM; *N* = 4; * *p* < 0.05; one-way ANOVA with Tukey’s test was used to assess significance.

**Figure 2 ijms-22-01491-f002:**
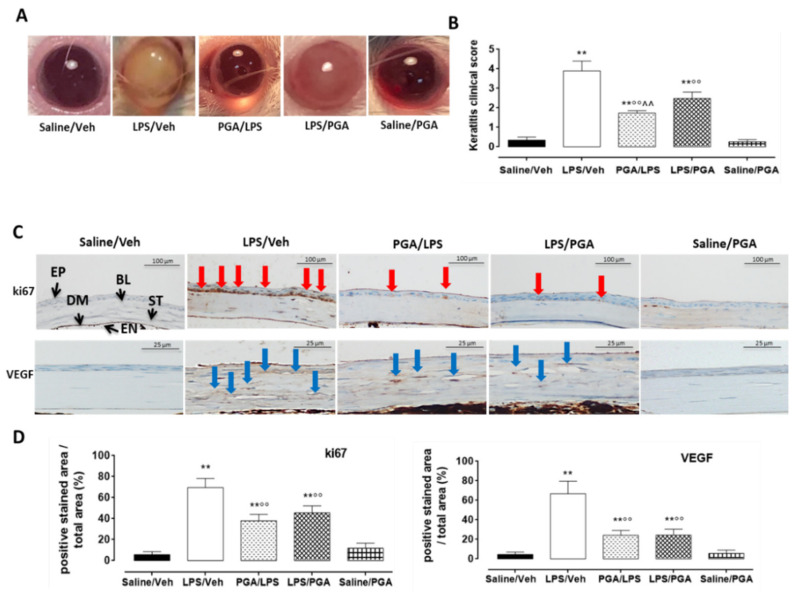
(**A**) Representative images of keratitis development in CD1 mice cornea. (**B**) Keratitis clinical scores. Positive keratitis was scored as > 1. Statistical significance was assessed by one-way ANOVA, followed by Tukey’s multiple comparison test of *N* = 5 mice/group. Saline/Veh = mice receiving intrastromal injection of 2 μL PBS; LPS/Veh = mice receiving intrastromal injection of 2 μg LPS dissolved in 2 μL PBS; PGA/LPS = mice administered with PGA 5 mg/kg per os 30 min before LPS intrastromal injection; LPS/PGA = mice administered for three days with PGA 5 mg/kg per os, starting from 30 min after LPS intrastromal injection; Saline/PGA mice administered for three days with PGA 5 mg/kg per os, starting from 30 min after PBS intrastromal injection. ** *p* < 0.01 vs. Saline/Veh; °° *p* < 0.01 vs. LPS/Veh; ^^ *p* < 0.01 vs. LPS/PGA. (**C**) Representative immunohistochemical images for ki67 and VEGF corneal staining. (**D**) Ki67 and VEGF protein levels were reported as percentage (%) ± S.D. of positive stained area/total area. Statistical significance was assessed by one-way ANOVA, followed by Tukey’s multiple comparison test of *N* = 5 histological observations for each individual/group. EP = epithelium; BL = Bowman’s layer; ST = stroma; DM = Descemet membrane; EN = endothelium; Veh = vehicle; PBS = phosphate buffered saline; LPS = lipopolysaccharide; PGA = *N*-palmitoyl-D-glucosamine; red arrows = Ki67 positive stain (scale bar: 100 µm); blue arrows = VEGF positive stain (scale bar: 25 µm). ** *p* < 0.01 vs. Saline/Veh; °° *p* < 0.01 vs. LPS/Veh.

**Figure 3 ijms-22-01491-f003:**
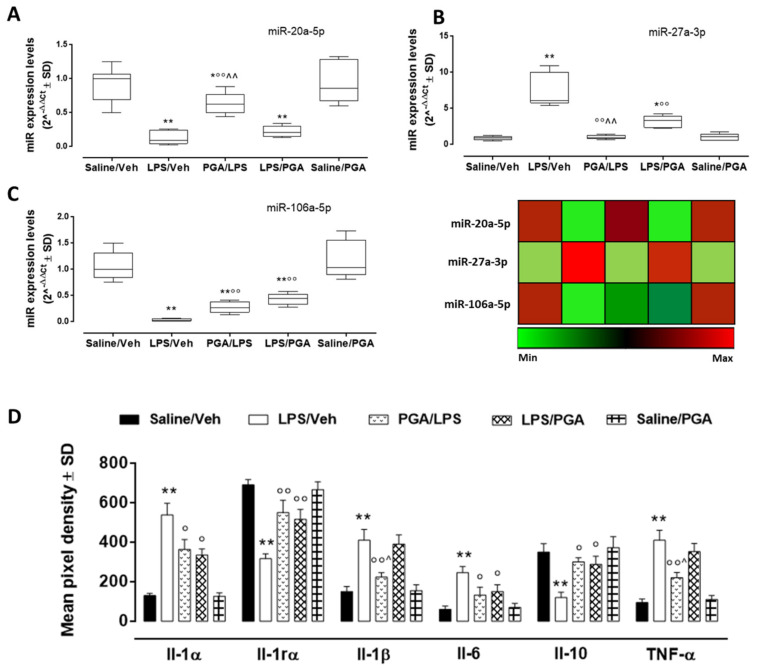
Dysregulated miRNA (**A**–**C**) and interleukin corneal levels (**D**). miR-20a-5p, miR-27a-3p and miR-106a expression levels, reported as median 2^^−ΔΔCt^ ± SD and depicted according to the color scale (red: upregulation; green: downregulation). Interleukins levels were reported as mean pixel density ± SD. Statistical significance was assessed by one-way ANOVA, followed by Tukey’s multiple comparison test of *N* = 3 measurements for each individual/group. Saline/Veh = mice receiving intrastromal injection of 2 μL PBS; PGA/LPS = mice administered with PGA 5 mg/kg per os 30 min before LPS intrastromal injection; LPS/Veh = mice receiving intrastromal injection of 2 μg LPS dissolved in 2 μL PBS; LPS/PGA = mice consecutively administered for three days with PGA 5 mg/kg *per os*, starting from 30 min after LPS intrastromal injection; Saline/PGA mice consecutively administered for three days with PGA 5 mg/kg *per os*, starting from 30 min after PBS intrastromal injection. * *p* < 0.05 and ** *p* < 0.01 vs. Saline/Veh; ° *p* < 0.05 and °° *p* < 0.01 vs. LPS/Veh; ^ *p* < 0.05 and ^^ *p* < 0.01 vs. LPS/PGA.

**Figure 4 ijms-22-01491-f004:**
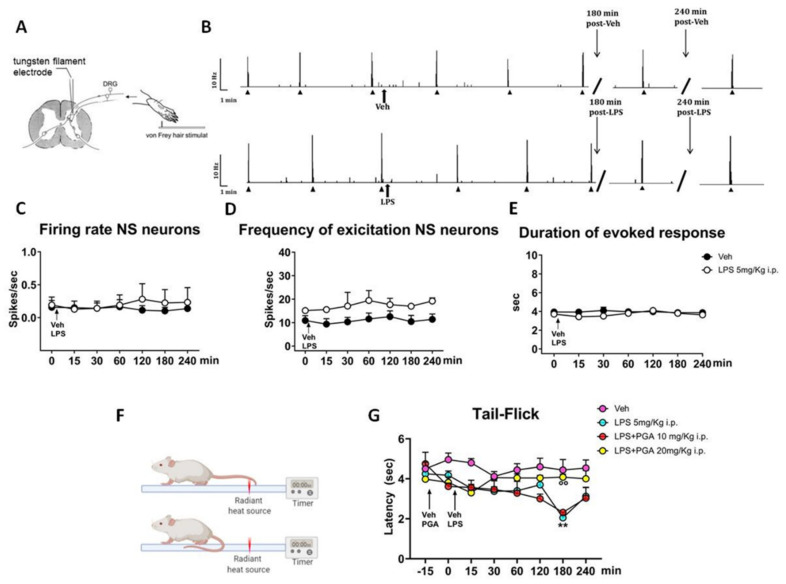
(**A**) Schematic diagram showing recording electrode placement for single-unit in vivo recording of NS neurons in spinal cord L4-L6 segments, after Von Frey stimulation on mouse hind paw. (**B**) Representative ratematers showing spontaneous and noxious-evoked activity of NS neurons recorded pre- and post- intraperitoneal injection of vehicle or LPS. (**C**) Spontaneous activity, (**D**) frequency and (**E**) duration of excitation of the evoked activity of NS neurons, in vehicle or LPS-injected mice. Each point represents the mean ± S.E.M of 3 different mice per group (one neuron recorded per each mouse). Two-way ANOVA for repeated measures followed by Tukey’s post hoc test was used for comparison between groups. (**F**) Representative diagram of tail-Flick test apparatus. (**G**) Effect of PGA (10 and 20 mg/kg), injected 15 min before LPS (5 mg/kg), on tail-flick latency in naïve mice (*N* = 5). Values are expressed as mean ± S.E.M. Two-way ANOVA for repeated measures followed by Tukey’s post hoc test was used for comparison between groups. ** *p* < 0.01 indicates statistical difference vs. Veh and °° *p* < 0.01 indicates statistical difference vs. LPS 5 mg/kg.

**Figure 5 ijms-22-01491-f005:**
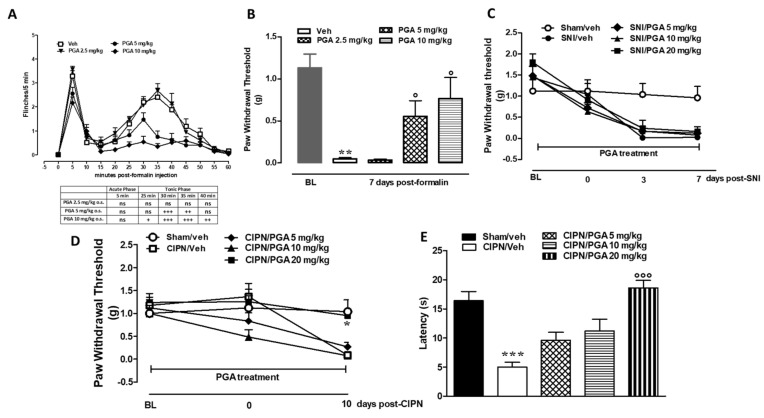
(**A**,**B**) Effect of PGA (2.5, 5 and 10 mg/kg os) in the formalin model of pain in mice. The total time of the nociceptive response was measured every 5 min and expressed in min and in repeated treatment on mechanical withdrawal threshold of the ipsilateral hind paw 7 days after formalin injection in mice. Modifications on withdrawal thresholds [expressed as applied force (g)] of hindpaw ipsilateral of SNI mice (**C**) after chronic injection of different doses of vehicle or PGA (5, 10 and 20 mg/kg, o.s.). Effect of PGA (2.5, 5, 10 and 20 mg/kg, os) on withdrawal thresholds and cold allodynia of hindpaw of CIPN mice (**D**,**E**). Data were expressed as mean ± SEM of mechanical withdrawal thresholds in grams. ^+^ (*p* < 0.05), ^++^ (*p* < 0.01) and ^+++^ (*p* < 0.001) indicate significant differences vs vehicle treatment in formalin injected mice. *(*p* < 0.05), ** (*p* < 0.01) and *** (*p* < 0.001) indicate significant differences vs veh/saline-treated mice, °(*p* < 0.05) °°° (*p* < 0.001) indicate significant differences vs. vehicle treatment (Two-way ANOVA followed by Tukey’s and Bonferroni post hoc test).

**Figure 6 ijms-22-01491-f006:**
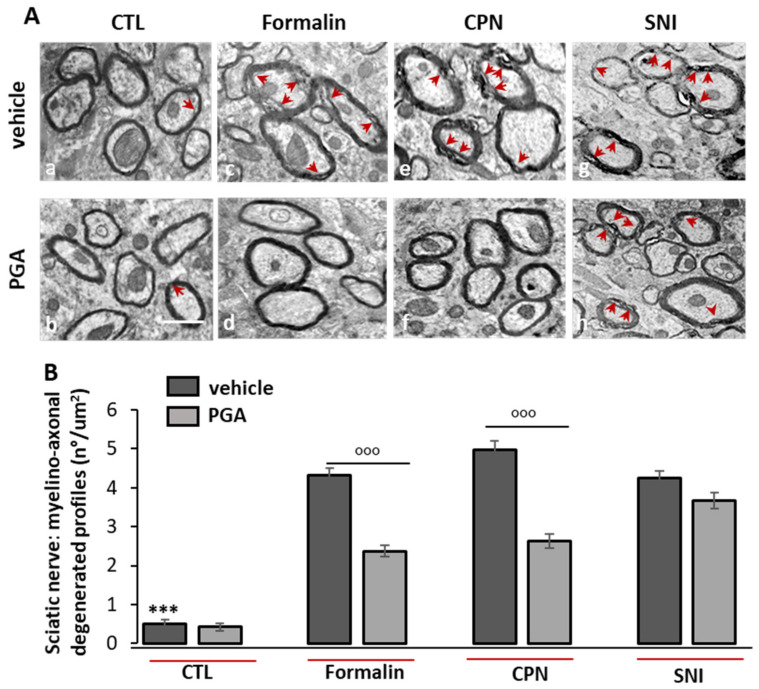
PGA (5–20 mg/kg o.s.) reinstated the ultrastructural changes of the impaired sciatic nerve in neuropathic mice. (**A**) Representative ultrastructural images of the myelin sheath showed that compared with control (a and b), PGA treatment restored the injured structure of sciatic nerve in formalin (c and d)- or CPN (e and f)-induced neuropathic mice, but not in SNI (g and h) mice, in which the myelin layers were damaged and detached from each other. Impaired fibers of sciatic nerve were indicated by red arrows [Scale bar: 1 μm]. (**B**) Bar graph showing the mean number of injured fibers for each treatment. Measure was expressed as means ± SEM of the number of myelino-axonal degenerated fibers/3micron^2^. One-way ANOVA followed Kruskal-Wallis test plus Dunn’s multiple comparison test was applied. *** *p* < 0.0001 is referred to vehicle + formalin or vehicle + CPN or vehicle + SNI. °°° *p* < 0.0001 is referred to PGA-treated formalin or CPN groups.

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
