# Peer review of "N-palmitoyl-D-glucosamine, A Natural Monosaccharide-Based Glycolipid, Inhibits TLR4 and Prevents LPS-Induced Inflammation and Neuropathic Pain in Mice"

_ijms, 2021, doi:10.3390/ijms22031491_

Round 1

Reviewer 1 Report

Please see the attached manuscript for comments. See below for supplementary figures comments.

Supplementary Figures comments:

(1) It looks like figure legend for figure S2 and S3 got exchanged. 

(2) Figure S3: Figure D legend is missing.

Author Response

We thank the Reviewer for the helpful suggestions. 

As indicated by the Reviewer, RAW264 has been replaced with RAW264.7.

As suggested by the Reviewer, more references were added to the sentence at the lines 49-50. 

TLR4 abbreviation was revised in the extended form. 

As suggested by the Reviewer, previous studies showing antiinflammatory effects of PGA were included in the Introduction (lines 69-72). 

Legends of supplementary figures have been revised.

The figure 5 has has benn moved in the related results section.

Reviewer 2 Report

Dear Authors, this topic covers the interesting research direction in the field and is enough nicely elaborated as the manuscript. However, I have some minor objections that should be addressed to improve the quality of the manuscript:

1) Introduction is too short and do not represent the topicality of the aim. Hypothesis covers almost one paragraph, but the basis of it is not completely proved; I would like to ask the authors to expand the Introduction, topicality and also the basis of the hypothesis with following clarified aim;

2) methods and Results - OK with an exception of Figure 6, where abbreviations are missed;

3) In Discussion part I would like to ask the authors precisely and in concentrate way to organize the Conclusions (what are intermixed with hypothesis and suggestions about the future work now...) in one paragraph;

4) Reference part enrolls 5 "old"sources from the previous century; please overthink they necessity and remove/replace them with more new ones as old References do not fit for this IF Journal and otherwise Yours really nice, complicated work!

Author Response

We thank the Reviewer for the helpful suggestions. 

1) A suggested by the Reviewer, the background has been improved. In particular, previous studies showing the antiinflammatory effetcs of PGA have been included. 

2) Abbreviations in results and legend of figure 6 have been revised.

3) The Conclusions of the study have been revised, accordingly with the suggestion given by the Revewer.  

4) References have been revised. However, few references regarding the methods used could not be replaced.